# The pro-resolving lipid mediator Maresin 1 ameliorates pain responses and neuroinflammation in the spared nerve injury-induced neuropathic pain: A study in male and female mice

Luísa Teixeira-Santos[1,2], Sandra Martins[3,4], Teresa Sousa[1,2], António Albino-Teixeira[1,2], Dora Pinho[1,2]*

1 Departamento de Biomedicina–Unidade de Farmacologia e Terapêutica, Faculdade de Medicina, Universidade do Porto, Porto, Portugal, 2 Centro de Investigação Farmacológica e Inovação Medicamentosa (MedInUP), Universidade do Porto, Porto, Portugal, 3 Serviço de Patologia Clínica, Centro Hospitalar e Universitário São João (CHUSJ), Porto, Portugal, 4 EPIUnit, Instituto de Saúde Pública, Universidade do Porto, Porto, Portugal

* dpinho@med.up.pt

## Abstract

Specialized pro-resolving mediators (SPMs) have recently emerged as promising therapeutic approaches for neuropathic pain (NP). We evaluated the effects of oral treatment with the SPM Maresin 1 (MaR1) on behavioral pain responses and spinal neuroinflammation in male and female C57BL/6J mice with spared nerve injury (SNI)-induced NP. MaR1, or vehicle, was administered once daily, on post-surgical days 3 to 5, by voluntary oral intake. Sensory-discriminative and affective-motivational components of pain were evaluated with von Frey and place escape/avoidance paradigm (PEAP) tests, respectively. Spinal microglial and astrocytic activation were assessed by immunofluorescence, and the spinal concentration of cytokines IL-1β, IL-6, IL-10, and macrophage colony-stimulating factor (M-CSF) were evaluated by multiplex immunoassay. MaR1 treatment reduced SNI-induced mechanical hypersensitivity on days 7 and 11 in both male and female mice, and appeared to ameliorate the affective component of pain in males on day 11. No definitive conclusions could be drawn about the impact of MaR1 on the affective-motivational aspects of pain in female mice, since repeated suprathreshold mechanical stimulation of the affected paw in the dark compartment did not increase the preference of vehicle-treated SNI females for the light side, during the PEAP test session (a fundamental assumption for PAEP's validity). MaR1 treatment also reduced ipsilateral spinal microglial and astrocytic activation in both sexes and marginally increased M-CSF in males, while not affecting cytokines IL-1β, IL-6 and IL-10 in either sex. In summary, our study has shown that oral treatment with MaR1 (i) produces antinociception even in an already installed peripheral NP mouse model, and (ii) this antinociception may extend for several days beyond the treatment time-frame. These therapeutic effects are associated with attenuated microglial and astrocytic activation in both sexes, and possibly involve modulation of M-CSF action in males.

**Data Availability Statement:** All data files are available from the figshare repository (DOI: 10. 6084/m9.figshare.22179991.v1).

**Funding:** This work was supported by University of Porto/Faculty of Medicine (https://sigarra.up.pt/fmup) and ESF – European Social Fund (https://ec.europa.eu/esf/home.jsp), through NORTE2020 – North Portugal Regional Operational Programme [NORTE-08-5369-FSE-000011-Doctoral Programmes – LTS' PhD fellowship], and by Fundação Grünenthal Portugal (https://www.fundacaogrunenthal.pt), Bolsa Jovens Investigadores em Dor 2018 – LTS). The funders had no role in study design, data collection and analysis, decision to publish, or preparation of the manuscript.

**Competing interests:** The authors have declared that no competing interests exist.

**Abbreviations:** CCI, chronic constriction injury; CNS, central nervous system; DHA, docosahexaenoic acid; EAE, experimental autoimmune encephalomyelitis; GFAP, glial fibrillary acidic protein; GPCR, G-protein-coupled receptor; Iba-1, ionized calcium binding adaptor molecule-1; LDB, light/dark box; LX, lipoxin; M-CSF, macrophage colony-stimulating factor; MaR, maresin; MWT, mechanical withdrawal threshold; NP, neuropathic pain; PD, protectin; PEAP, place escape/avoidance paradigm; Rv, resolving; SCI, spinal cord injury; SNI, spared nerve injury; SPM, specialized pro-resolving mediator.

## Introduction

Neuroinflammation, a localized form of inflammation occurring within the peripheral and central nervous system, is critically involved in neuropathic pain (NP) pathophysiology. The nerve injury-induced neuroinflammatory response is primarily characterized by activation of glial cells and production of inflammatory mediators (e.g. cytokines and chemokines) [1, 2]. Mounting evidence has revealed changes in morphology, number, and function of both microglia and astrocytes after nerve damage, resulting in peripheral and central sensitization [3, 4].

Ideally, the inflammatory response is a self-limited process that culminates with complete resolution, enabling the restoration of homeostasis. In fact, the resolution of inflammation is now recognized as an active and coordinated process, governed by a group of endogenous chemical mediators, termed as specialized pro-resolving mediators (SPMs) [5, 6]. They include lipoxins (LXs), produced by metabolism of arachidonic acid, and maresins (MaRs), protectins (PDs), and resolvins (Rvs), which are biosynthesized from n-3 polyunsaturated fatty acids (eicosapentaenoic acid (EPA), docosahexaenoic acid (DHA), or docosapentaenoic acid (DPA)) [6, 7]. By activating G-protein-coupled receptors (GPCRs), SPMs have several pro-resolving and anti-inflammatory actions, such as cessation of immune cell infiltration, counterregulation of proinflammatory factors, induction of anti-inflammatory mediators, stimulation of efferocytosis (phagocytic clearance of apoptotic cells), antimicrobial killing, and tissue regeneration [6–8]. Failed resolution can lead to uncontrolled/persistent inflammation, which has been associated with multiple diseases [7], including NP [9, 10]. Therefore, strategies that aim to stimulate or accelerate/intensify the resolution of inflammation (in contrast to traditional strategies that mainly focused on suppressing, blocking, or inhibiting proinflammatory mediators) have emerged as promising therapeutic approaches to treat inflammation-associated conditions [7, 8, 11]. In experimental NP models, treatment with SPMs have consistently resulted in attenuation of nociceptive behaviors and reduction of nerve injury-induced neuroinflammation. In fact, SPMs are now acknowledged as a new class of non-immunosuppressive and non-opioid analgesic drugs [9, 12, 13].

Maresin 1 (MaR1; 7R,14S-dihydroxy-4Z,8E,10E,12Z,16Z,19Z-DHA) is a DHA-derived SPM biosynthesized by macrophages [14], with potent pro-resolving and anti-inflammatory actions [15]. To date, MaR1 has been shown to activate two classes of receptors: the leucine-rich repeat containing G protein-coupled receptor 6 (LGR6), a plasma membrane GPCR [16], and the retinoic acid-related orphan receptor α (ROR-α), a nuclear receptor [17]. MaR1 was also shown to act as partial agonist of recombinant human leukotriene B$_4$ receptor (BLT1) [16, 18]. Importantly, accumulating evidence has revealed that different cells of the central nervous system (CNS) express several SPM receptors, including LGR6 and ROR-α, which suggest that MaR1 may be able to modulate neuroimmune processes by targeting neurons and/or glia [19].

MaR1 antinociceptive actions have been demonstrated in diverse experimental pain models, including NP models (spinal nerve ligation (SNL) [20], chronic constriction injury (CCI) [21], radicular pain [22, 23], and vincristine-induced NP [15]), models of inflammatory pain induced by capsaicin [15], carrageenan, and Complete Freund's Adjuvant (CFA) [24], and post-operative pain induced by tibial bone fracture (fPOP) [25]. None of these studies, however, included female rodents, despite accumulating evidence uncovering robust differences between sexes in the neuroimmune modulation of pain, which may affect the responsiveness of male and female subjects to analgesic drugs in preclinical and clinical contexts [26, 27]. The possible existence of sexual dimorphism in the analgesic efficacy of some SPMs should not be overlooked, as recently evidenced by Luo *et al.*, who reported that RvD5 showed antinociceptive actions in male, but not female, mice with neuropathic and inflammatory pain [28]. Furthermore, although the oral route is highly relevant due to its unique advantages in clinical

settings, the effects of orally administered MaR1 have not been evaluated in experimental pain models.

In the present study, aiming to further explore the therapeutic potential of MaR1 in peripheral NP by addressing possible sexual dimorphisms and pursuing administration routes with higher translational relevance, we assessed the effects of oral treatment with MaR1, using a voluntary oral intake protocol, on behavioral responses and spinal inflammatory parameters, in male and female mice with spared nerve injury (SNI)-induced NP.

## Material and methods

### Animals

Forty male (n = 20) and female (n = 20) C57BL/6J mice (specific pathogen-free; 7 week-old upon arrival), purchased from Charles River (Lyon, France), were used in this study. Animals were housed in same-sex groups of two or four, under controlled temperature (21–24˚C) and humidity (45–55%) conditions, in a 12 h light/dark cycle (lights on at 8:00 AM). Food and water were provided *ad libitum*, and nesting material and cardboard tunnels (one per cage) were used as environmental enrichment.

All experimental procedures were carried out in compliance with the national (Portuguese Decree-Law 113/2013) and international (European Directive 2010/63/EU) guidelines for experimental research in animals, and specific guidelines for the study of pain (IASP Guidelines for the Use of Animals in Research, [29]), with the approval of the local Animal Welfare Committee and the competent national authority, *Direção-Geral de Alimentação e Veterinária* (DGAV–Ref. 0421/000/000/2020).

Monitoring and assessment of animal welfare was performed using a clinical observation/scoring system described in pages 30–31 of the "Classification and reporting of severity experienced by animals used in scientific procedures: FELASA/ECLAM/ESLAV Working Group report" [30].

### Experimental design and drug administration

Experimental protocol, including research question, key design features, and analysis plan, was prepared before the study, but was not registered.

Mice were allowed to acclimatize to the animal facility for a minimum of 7 days, followed by a period of habituation to handling by the experimenter in the testing room.

All animals were subjected to SNI surgery (set as "day 0") after being randomly allocated within sex to one of two treatment groups: SNI-MaR1 (n = 20) and SNI-vehicle (n = 20). MaR1 was purchased from Cayman Chemical (Cat. No. 10878; supplied as a solution containing 100 μg/mL in 100% ethanol) and was divided in single-use aliquots. Depending on the treatment group, MaR1 (50 μg/kg) or vehicle were administered once daily on days 3 to 5 (Fig 1), by voluntary oral intake, using a protocol developed by our research group [31]. Due to their neophobic nature, mice needed a previous 3-day habituation period, during which they were placed in individual cages (which were used for the same animal throughout the entire drug-administration protocol) once daily (for 10 min), and presented with a 60-μL drop of strawberry jam (*Doce Froiz Morango extra*, Dulces Y Conservas Helios S.A., Spain). On each treatment day, an aliquot of MaR1 per animal was thawed and further prepared by evaporating the ethanol with a gentle stream of nitrogen gas until a final volume of *ca.* 5 μL, and then adding 6 μL of sterile 0.1 M phosphate buffered saline (PBS) pH 7.4, for immediate use. The vehicle solution consisted of 6 μL of PBS *plus* 5 μL of ethanol. Each mouse was placed into the respective cage and presented with a 60-μL drop of jam containing the appropriate solution. Further details regarding the voluntary oral intake protocol (such as data analysis on latency to

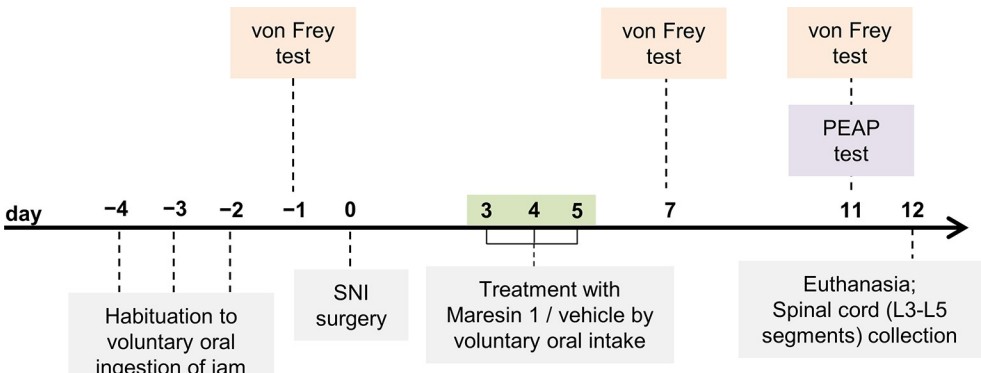

**Fig 1. Schematic representation of the experimental design.** Male and female mice were randomly allocated within sex to SNI-MaR1 or SNI-vehicle groups and habituated for 3 consecutive days to voluntarily ingest a strawberry jam. SNI surgery day was set as day 0. MaR1 or vehicle were administered once daily, from days 3 to 5, by voluntary oral intake. Behavioral pain responses were evaluated with either the von Frey test (before surgery–baseline, and on days 7 and 11) or the PEAP test (on day 11). Mice were euthanized on day 12.

ingestion and discussion of its strengths and weaknesses) can be found in our previous publication [31].

The treatment scheme selected aimed to increase the translational potential of our findings, by evaluating the effects of MaR1 treatment when administered after activation and initial development of SNI-induced inflammatory processes and of mechanical hypersensitivity [32, 33], and was similar to the one used by Gao *et al.*, who reported that daily intrathecal administration of MaR1 on postsurgical days 3, 4 and 5 attenuated pain hypersensitivity and neuroinflammation in a NP rat model [20]. The choice of the 50 µg/kg MaR1 dose was based on studies by Moreno-Aliaga's group, which have reported the administration of MaR1 by oral delivery (through intragastric gavage) [34–38].

The sensory-discriminative component of pain was evaluated with the von Frey test, before surgery (baseline), and on days 7 and 11 post-SNI, whereas the place escape/avoidance paradigm (PEAP) test was performed only once, on day 11, for the evaluation of the affective-motivational component of pain. Each mouse was subjected to either von Frey or PEAP protocols. Euthanasia and tissue collection were performed on day 12 (Fig 1). The behavioral pain assessment and tissue collection timeline was designed to evaluate the ability of a "short" MaR1 treatment to both reverse the already initiated SNI-induced inflammatory processes and mechanical hypersensitivity, and prevent the NP chronification.

Based on a biologically relevant effect size-to-standard deviation ratio *ca.* 1.8 for the behavioral evaluation of pain, we estimated a sample size of 6 experimental units per group, using InVivoStat power analysis module, for a power of 80% and a significance level of 5%. In this study, each mouse constituted an experimental unit, since mice could be randomly allocated to either treated or vehicle group, even though they were housed in groups (the drug *vs.* vehicle treatment was individually administered by voluntary ingestion). Since each mouse was subjected to only one test (either von Frey or PEAP), and 6 sets of 4 animals (1 vehicle-treated SNI male + 1 vehicle-treated SNI female + 1 MaR1-treated SNI male + 1 MaR1-treated SNI female) were necessary for each test, a total of 2 × 24 animals (2 × 12 males and 2 × 12 females) was determined as necessary. To further prevent unnecessary use of animals, we decided beforehand to perform interim analyses of the behavioral results after 4 sets of 4 animals, in order to ascertain whether or not sex differences were apparent (and, subsequently decide to complete, or not, the remaining sets [39]). For the von Frey mechanical threshold determination, this

interim analysis revealed no sex-based differences, and we therefore pooled the results from male and female animals (thereby obtaining n = 8, in excess of n = 6 required by the power analysis) and did not use the two additional 4-animal sets that would have been needed for the quantification of sex-based differences. On the other hand, sex effects were evident in the PEAP test, and therefore in this case the 6 initially-prescribed sets of 4 animals were required. Therefore, a total of 10 animal sets of 4 animals were used. Upon euthanasia, L3-L5 spinal cord segments were collected after *in situ* fixation in half of the animal sets (used for the assessment of microglial and astrocytic activation, by immunofluorescence), or as fresh tissue in the other half (used for quantification of cytokines IL-1β, IL-6, IL-10, and macrophage colony-stimulating factor (M-CSF), by multiplex assays).

## Spared nerve injury surgery

NP was induced in 11 weeks-old mice by the SNI model [32]. Mice were anesthetized with an intraperitoneal (i.p.) injection of ketamine (75 mg/kg) and medetomidine (1 mg/kg). Absence of a pedal reflex following a firm toe pinch was monitored to ensure appropriate anesthetic depth. After disinfecting the operative field, a skin incision was made in the longitudinal direction proximal to the left knee. Then, a section was made through the *biceps femoris* muscle, exposing the sciatic nerve and its three peripheral branches. A tight ligation of the tibial and common peroneal nerves was performed with a 6.0 silk thread (Fine Science Tools), and a 1–2 mm distal section of the nerve stump was removed. The sural nerve was preserved by avoiding nerve stretching or contact with surgical tools. Muscle and skin were closed in two distinct layers with 5.0 silk suture (Silkam, B. Braun Medical). Sedation was reversed with atipamezole (2 mg/kg, subcutaneous injection). The surgical site was examined in each animal after euthanasia, to confirm that the sural nerve was intact, and no nerve regeneration had occurred. Two exclusion criteria were set: (1) animals that did not develop mechanical hypersensitivity after SNI, and (2) animals that, upon *post-mortem* inspection, showed severed/damaged sural nerve or signs of tibial/common peroneal nerve regeneration.

## Behavioral tests

Behavioral testing was conducted during the light phase by a female experimenter. Animals were allowed to habituate to the testing room for at least 1h30 before the beginning of experimental procedures.

**von Frey test.** Mechanical withdrawal threshold (MWT) was assessed with the von Frey test, using the "ascending stimulus" method [40]. Mice were placed in transparent acrylic cylinders (inner diameter: 8.4 cm; height: 7 cm), wrapped in a grey plastic strip to darken the inside, on an elevated perforated metal platform. Animals were habituated to the apparatus for 30 min during five days before the first test and, on each test day, for at least 15 min before beginning the procedure. A series of 9 calibrated von Frey filaments (0.008, 0.02, 0.04, 0.07, 0.16, 0.4, 0.6, 1 and 1.4 g; Stoelting Co., Wood Dale, IL, USA) was perpendicularly applied to the sural nerve skin territory of the ipsilateral hind paw until bending. Ten stimuli were applied with each filament over a total period of 30 s (approximately 2 s per stimulus), starting with the lowest one (0.008 g) and proceeding in ascending order until the MWT (determined as 2 positive responses in 10 stimulations) was reached [32, 41]. A sudden paw lifting, flinching, guarding, or licking was considered a positive response.

**Place escape/avoidance paradigm test.** The PEAP test, which creates a conflict between an innately aversive light compartment and an aversive noxious mechanical stimulation to the injured paw in a dark compartment, aims to evaluate the affective/emotional component of pain [42–44].

The test was performed in a testing box with 2 compartments interconnected through a hole (5 × 5 cm). One compartment was transparent, and the other was opaque black (15 × 15 × 20 cm each). The box was positioned on top of a perforated metal floor and a light bulb (330 lm) was placed *ca.* 5 cm above the uncovered transparent chamber to further illuminate it, while the dark chamber was covered with a black lid. Two days before the assay, mice were habituated to the perforated metal floor once for 30 min. On the testing day, animals were placed within the light side of the testing box and allowed unrestricted movement between the two areas throughout the test period (30 min). A suprathreshold (1 g) von Frey filament was applied for approximately 1 s, at 15-s intervals, to the lateral plantar surface of the hind paw [45]. The ipsi- or contralateral paw was stimulated when the mouse was in the dark or light chamber, respectively. Thus, each mouse was given the choice to move to the light chamber in order to escape/avoid the noxious stimulation of the ipsilateral hind paw in the dark side. Since mice naturally prefer the dark side, the time spent in the light chamber is considered as a measure of the aversion to noxious stimulation of the injured hind paw relative to aversion to the bright chamber [43, 45]. Test sessions were recorded using a video camera (Sony Handy-Cam HDR-CX240E). The box was cleaned with 35% ethanol between subjects to eliminate potential olfactory cues.

The location of the animal (light or dark side) throughout the test period and the number of crossings from one side to the other were assessed using Solomon Coder Software (Version beta 17.03.22; https://solomon.andraspeter.com/). To evaluate the escape/avoidance behavior of each mouse, in addition to the total time spent in the light chamber and the total number of midline crossings, total time was binned into six 5-min intervals, and both percentage of time and accumulated percentage of time spent in the light chamber were further calculated for each time bin. The shift in preference (difference in % of time spent in the light chamber between the last and first 5-min time bins) was also calculated [44–48].

## Immunofluorescence

Immunofluorescence staining for ionized calcium binding adaptor molecule 1 (Iba-1) and glial fibrillary acidic protein (GFAP) were performed to assess microglial and astrocytic activation, respectively.

Animals were deeply anesthetized with sodium pentobarbital (100 mg/kg, i.p. injection) and perfused through the ascending aorta with 0.1 M PBS pH 7.4 followed by 4% paraformaldehyde, for *in situ* fixation of tissues. Spinal cord L3-L5 segments were collected, post-fixed in the same fixative for 3 h, and transferred to a 30% sucrose solution with 0.1% sodium azide at 4°C. The contralateral side was identified with a small cut in the ventral horn. Spinal cord segments were transversely cut in 30 μm sections (4 series) using a freezing microtome (Leica CM 1325). Sections were stored at −20°C in a cryoprotectant solution of 30% (m/v) sucrose dissolved in phosphate buffer 0.1 M and 30% (v/v) ethylene glycol, until being used in immuno-fluorescence assays.

Free-floating spinal cord sections (one series of each animal) were washed in 0.1 M PBS, treated with 1% sodium borohydride in PBS for 30 min, washed again in PBS and PBS with 0.3% Triton X-100 (PBST), and incubated for 2 h with a blocking solution containing 0.1 M glycine and 10% normal horse serum (NHS, Gibco, Cat. No. 16050130) in PBST. Then, the sections were incubated with primary rabbit polyclonal antibodies against Iba-1 (1:1000; Fuji-film Wako, Cat. No. 019–19741, RRID:AB_839504; antigen: synthetic peptide—Iba1 C-termi-nal sequence; batch number: PTH4470) or GFAP (1:500; Dako/Agilent, Cat. No. Z0334, RRID:AB_10013382; immunogen: GFAP isolated from cow spinal cord; batch number: 20071831), diluted in PBST with 2% NHS, for 3 overnights at 4°C or 1 overnight at room

temperature, respectively. Sections were washed in PBST and then incubated for 1 h with Alexa Fluor-594 donkey anti-rabbit polyclonal secondary antibody (1:1000; Invitrogen, Cat. No. A21207, RRID:AB_141637; batch number: 2066086). Finally, after repeated washing with PBST and PBS, sections were mounted on gelatin-coated slides, dried overnight at 4˚C and coverslipped with Prolong Gold antifade reagent with DAPI (Invitrogen Ltd., Cat. No. P36941). In order to establish the specificity of the immunostaining, negative controls were performed by replacing the incubation with primary antibody solutions by PBST with 2% NHS only.

Images of the stained sections were acquired using 2.5 × and 10 × objectives, using a fluorescence microscope (Axio Imager.Z1, Zeiss, Germany), through an AxioCam MRm digital camera with AxioVision 4.6 software (Carl Zeiss MicroImaging GmbH), under the same image acquisition settings. Using the open-source software Fiji [49], Iba-1 and GFAP fluorescent intensities were measured in images captured with a 10 × objective. Images were first converted to 8-bit grayscale and mean grey values were measured within a rectangle region of fixed dimensions (266 × 159 μm) comprising the medial two thirds of the dorsal horn (laminae I-III) [50]. The ipsilateral and contralateral sides were evaluated, and the average ipsilateral/contralateral ratio was calculated to normalize the data. Samples severely damaged in the region of interest were not analyzed and only animals with a minimum of 5 sections available were included in the statistical analysis.

## Multiplex assay

After being deeply anesthetized with i.p. sodium pentobarbital (100 mg/kg), mice were transcardially perfused through the ascending aorta with ice-cold phosphate-buffered saline (PBS) 0.1 M pH 7.4. Spinal cords (L3-L5 segments) were collected, snap-frozen in liquid nitrogen and stored at −80˚C. Frozen samples were homogenized in cold lysis buffer (300 μL/10 mg of tissue) composed by 50 mM Tris-HCl pH 7.4, 150 mM NaCl, 1% IGEPAL, and protease (cOmplete, Mini Protease Inhibitor Cocktail, Roche) and phosphatase (PhosSTOP Phosphatase Inhibitor Cocktail, Roche) inhibitors. Tissues were sonicated in a bath sonicator (3 × 5 min, at 4˚C) and then centrifuged (12 000 × g, 4˚C, 10 min). Supernatants were aliquoted and stored at −80˚C until multiplex assay was performed. Total protein concentration was quantified by the Bradford method using bovine serum albumin as a standard.

Proinflammatory IL-1β and IL-6, anti-inflammatory IL-10, and M-CSF, which is involved in the regulation of microglial development, proliferation, and maintenance [51], were simultaneously quantified in each sample using the Milliplex MAP Mouse Cytokine/Chemokine Magnetic Bead Panel kit (Merck Millipore), according to the manufacturer's protocol, on a Luminex 200™ xMAP™ Technology analyzer (Luminex Corp., TX, USA) which uses a dual-laser flow cytometry-based technology. Before the assay, the protein concentration of each sample was adjusted to 2 mg/mL with lysis buffer. In brief, this immunoassay involves incubating the protein extract with fluorescent-coded magnetic beads pre-coated with capture antibodies, followed by biotinylated detection antibodies and streptavidin-phycoerythrin conjugate. Raw data analysis (mean fluorescence intensity) was performed using a standard five parameter logistic (5-PL) curve fit created by the Luminex xPONENT Software (version 3.1). Results are presented as picograms per mg of total protein content.

## Statistical analyses

Statistical analyses were performed with InVivoStat 4.3.0 (Cambridge, UK), a statistical software package that uses R as its statistics engine [52]. Assumptions for the use of parametric analysis were ascertained using diagnostic plots: *Normal probability plot*, for the assumption of

normal distribution of the residuals, and *Predicted vs. Residuals plot*, for the assumption of homogeneity of the variance. Log10 transformation was applied to von Frey MWT, since the original values fulfilled neither of the required assumptions.

Results from the PEAP test (total time in light, shift in preference, total number of crossings), immunofluorescence and multiplex quantifications were analyzed with single measure parametric analysis (ANOVA), with factors *Treatment* (MaR1, vehicle) and *Sex* (male, female). Log10-transformed MWTs (von Frey testing) and time spent in the light chamber in each 5-min bin (PEAP testing) were analyzed with repeated measures parametric analysis (mixed model), with *Treatment* and *Sex* factors, and *Time* as the repeated factor (corresponding to *Day* in the analysis of von Frey results; and *5-min time bins*, in the analysis of data from the PEAP test). Planned comparisons were further conducted to separately assess between- and within-groups' differences, using the least significant difference (LSD) test and corrected with Holm's procedure for multiple comparisons. Correlation analyses were performed with Pearson's correlation coefficient.

In both single and repeated measures procedures, when no *Sex* main effects or interactions were detected, data from males and females within the same treatment group were pooled [53, 54]. *In vivo* procedures were divided into several mini-experiments, or blocks, such that each block contained an integer multiple of the 4-mice set described in *Experimental design and drug administration* section, and statistical analysis accounted for the blocking factor, whenever a blocking effect was detected.

Graphic illustrations were created with GraphPad Prism 9.2.0 for Windows (San Diego, California, USA). Results are expressed as predicted means (95% confidence interval, CI), unless otherwise stated. The significance level was set at 5%.

## Results

### Effects of MaR1 treatment on SNI-induced mechanical hypersensitivity

The von Frey test was performed to assess the effects of MaR1 treatment on SNI-induced mechanical hypersensitivity. Since no significant differences between sexes were found, the final analysis did not include the factor *Sex* (as described in *Statistical analyses* section). One animal did not develop mechanical hypersensitivity after SNI and therefore was excluded from the study.

Repeated measures parametric analysis revealed significant main effects of treatment (F(1, 13) = 5.00, *P* = 0.044) and time (F(2, 26) = 93.06, *P* < 0.0001). MWTs did not differ between treatment groups at baseline (SNI-MaR1 *vs.* SNI-veh, *P* = 0.91). After surgery, a significant decrease of the MWTs was observed in SNI-mice (for both SNI-veh and SNI-MaR1: BL *vs.* day 7, *P* = 0.0002; BL *vs.* day 11, *P* = 0.0002; Fig 2; S1 Fig), confirming the development of SNI-induced mechanical hypersensitivity. Treatment with MaR1 increased log10-transformed MWTs by 18.4% (SNI-MaR1 *vs.* SNI-veh, *P* = 0.037) on day 7, and by 20.2% (SNI-MaR1 *vs.* SNI-veh, *P* = 0.020) on day 11.

### Effects of MaR1 treatment on escape/avoidance behavior after SNI

To dissociate the complex and multidimensional pain experience of SNI-mice, the PEAP test was used on day 11 as a measure of the emotional/affective component of NP [43]. An underlying assumption of this test is that mice naturally prefer the dark compartment. In our study, all individuals spent more than 50% of the overall time in the dark chamber (Fig 3C), which confirms animals' innate preference, although large inter-individual variability was observed in individual time bins (S2 and S3 Figs), which has been reported in other rodent pain models as well [44, 55]. Upon repeated suprathreshold mechanical stimulation of the affected paw in

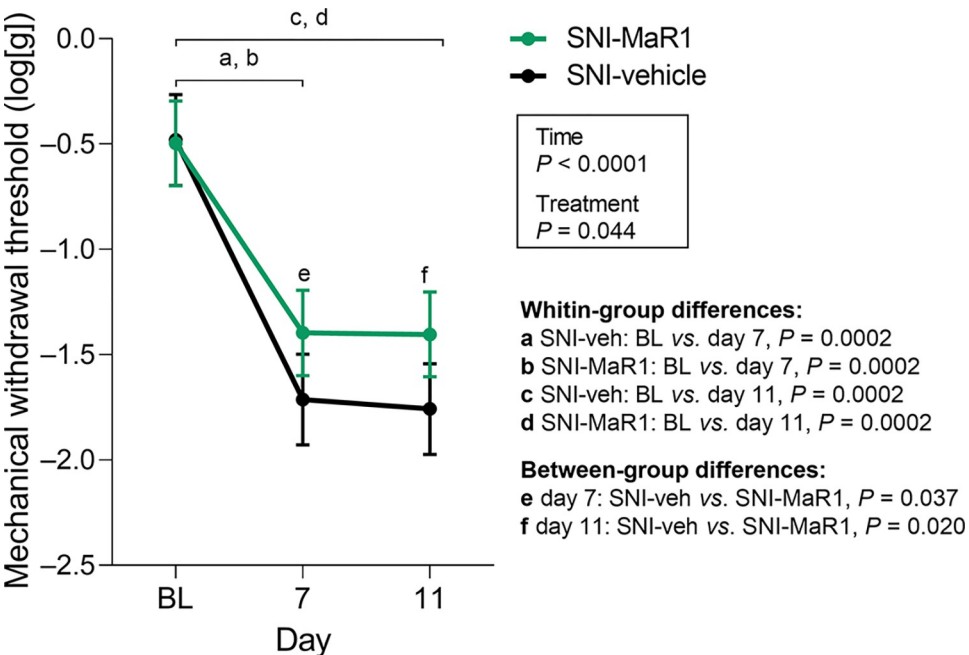

**Fig 2. MaR1 treatment reduced spared nerve injury (SNI)-induced mechanical hypersensitivity in the ipsilateral hind paw.** Mechanical withdrawal thresholds (MWTs) were assessed before surgery (baseline, BL), and on days 7 and 11. Data were analyzed with repeated measures parametric analysis (mixed model), with factors *Treatment* and *Time*. Holm's procedure was used to correct for multiple comparisons. Mean predicted values of log10-transformed values (95% CI) from the analysis are presented. *P*-values for statistically significant main effects/interactions and planned comparisons are indicated. SNI-veh, n = 7; SNI-MaR1, n = 8. A female mouse from the SNI-vehicle group did not develop mechanical hypersensitivity after SNI and was therefore excluded from the study. Individual log10-transformed values are presented in S1 Fig.

the dark compartment, SNI-mice are expected to gradually increase their preference for the light side during the test session.

Although some studies have used the total time spent in light as primary outcome measure in the PEAP test [46, 56–60], other parameters, more informative about the behavior throughout the test, may also be used as indicators of escape/avoidance, to better reflect the associative learning process that underlies this assay. In order to get an understanding of changes in mice behavior during the 30-min test period, we started our analysis by plotting the time spent in light on two graphs: individual and accumulated time spent in the light area for each 5 min bin of PEAP testing (Fig 3A and 3B). Visual inspection of the graphs shows that the behavioral pattern of vehicle-treated SNI male and female mice were qualitatively different. Indeed, when analyzing the individual or accumulated time in light, the mixed model analysis revealed not only a statistically significant main effect of *Time bin* (individual time in light: F(5, 100) = 3.93, P = 0.003; accumulated time in light: F(5, 100) = 2.60, P = 0.030), but also a *Sex × Time bin* interaction (individual time in light: F(5, 100) = 2.65, P = 0.027; accumulated time in light: F(5, 100) = 3.64, P = 0.005).

The time spent in light during the course of the test was significantly increased only in vehicle-treated SNI male mice (Fig 3A and 3B). In this subgroup, the differences between the last and the first two time bins were statistically significant (individual time in light: 5-min bin *vs.* 30-min bin, P = 0.026, 10-min bin *vs.* 30-min bin, P = 0.032; accumulated time in light: 5-min bin *vs.* 30-min bin, P = 0.006, 10-min bin *vs.* 30-min bin, P = 0.007). In fact, male vehicle-treated SNI mice displayed a 16% shift in preference by the end of experiment (Fig 3D), which

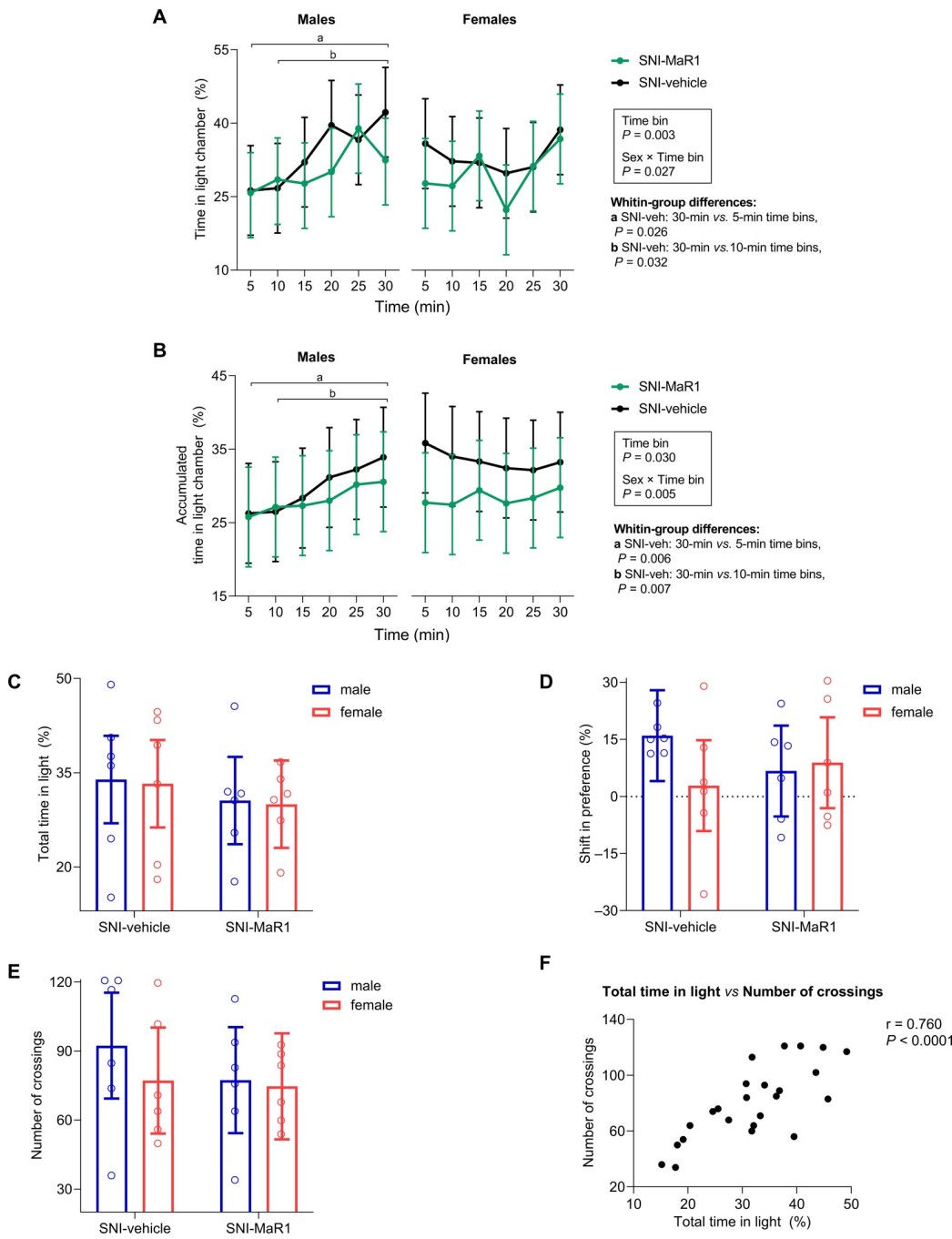

**Fig 3. Effects of MaR1 treatment in the motivated pain escape/avoidance behavior, on day 11 after spared nerve injury (SNI).** The timeline of preference of SNI animals throughout the 30-min test period was different in male and female mice, and only SNI-vehicle male mice significantly shifted their preference to the light side, as assessed by inspection of the time spent in the light chamber (%) **(A)**, and the accumulated time in light **(B)**, in time bins of 5 min. No statistically significant differences were found in the total time spent in the light chamber (%) **(C)**; the shift in preference (difference in % of time spent in the light chamber between the last and first 5-min time bin) **(D)**; or the total number of crossings between chambers **(E)**. Nevertheless, the escape/avoidance behavior was attenuated in male MaR1-treated mice as compared to their vehicle-treated controls. Data from graphs A and B were analyzed with repeated measures parametric analysis (mixed model), with factors *Treatment*, *Sex* and *Time bin*, and are presented as mean predicted values (95% CI) (individual values are shown in S2 and S3 Figs), whereas data from graphs C-E were analyzed with single measures ANOVA, with factors *Treatment*, and *Sex*, and are presented as individual values and mean predicted values (95% CI). In both cases, Holm's procedure was used to correct for multiple comparisons. *P*-values for statistically significant main effects/interactions and planned comparisons are indicated. ♂-SNI-veh, n = 6; ♂-SNI-MaR1, n = 6; ♀-SNI-veh, n = 6;

♀-SNI-MaR1, n = 6. A positive correlation was found between the total time in light and the total number of centerline crossings (**F**). Pearson's correlation coefficient analysis included all animals (N = 24).

demonstrates that the aversiveness of mechanical stimulation of the injured paw was higher than the aversiveness of the light chamber. In contrast, some female SNI-vehicle mice spent more time in the light chamber in the beginning of the assay (particularly when compared to MaR1-treated females, although the difference was not statistically significant; S2B and S3B Figs), and then increased their preference for the dark chamber (Fig 3D), which contradicts the PEAP assumption.

Although conclusions about MaR1 effects on SNI-females are hindered by the erratic behavior displayed by vehicle-treated SNI female mice, all vehicle-treated SNI males exhibited, as expected, an increase in light preference over the course of the test, which validates the use of the PEAP test in male SNI-mice under our test conditions. No statistically significant differences were found between vehicle- and MaR1-treated mice on the degree of aversion to the noxious stimuli, as assessed by the analysis of the time spent in light throughout the 30-min period of testing (Fig 3A and 3B), the total time in light (Fig 3C), and the shift in preference (Fig 3D). We also measured the total number of crossings from one side of the testing box to the other side, which has been considered as an index of general motor activity in the PEAP test [44, 46, 61]. Again, there were no significant differences between male and female SNI-mice treated with MaR1 or vehicle. Thus, locomotor activity/exploration are unlikely to have confounded the results. Additionally, a global correlation analysis of our results showed that the number of crossings was positively correlated with the total time in light (number of crossings *vs.* total time in light: correlation coefficient = 0.760; *test statistic* = 5.478; $P < 0.0001$; Fig 3F).

Nevertheless, although the mean behavior of male SNI-mice from both groups was relatively similar during the first time bins (which can be explained by the fact that animals are still "learning" that the dark chamber is associated with increased unpleasantness; Fig 3A and 3B), statistically significant differences between the time spent in light in the last and the first time bins, as already mentioned, were observed in the vehicle-treated, but not MaR1-treated, SNI-mice. Indeed, although the difference did not reach statistical significance, male mice treated with MaR1 developed a 58% lower shift in preference during the test than their vehicle-treated male counterparts (Fig 3D). Taken together, the behavior of MaR1-treated males appears to reflect an attenuation of the escape/avoidance behavior induced by mechanical stimulation of the injured paw, which is compatible with amelioration of the affective component of pain.

## Effects of MaR1 treatment on SNI-induced spinal microglial and astrocytic activation

To assess whether treatment with MaR1 affected glial activation after SNI surgery, immunoreactivity for Iba-1 (marker of microglial activation) and GFAP (marker of astrocytic activation) were evaluated on the dorsal horn of spinal cords collected on day 12.

Since no significant differences between sexes were found, the final analysis did not include the factor *Sex* (as described in *Statistical analyses* section). When compared to vehicle-treated mice, treatment with MaR1 reduced Iba-1 staining intensity by 11.1% ($F_{(1, 16)} = 5.25$, $P = 0.036$; Fig 4A and 4C), and reduced GFAP staining intensity by 10.1% ($F_{(1, 16)} = 4.54$, $P = 0.049$; Fig 4B and 4D), which indicates attenuation of microglial and astrocytic activation, respectively.

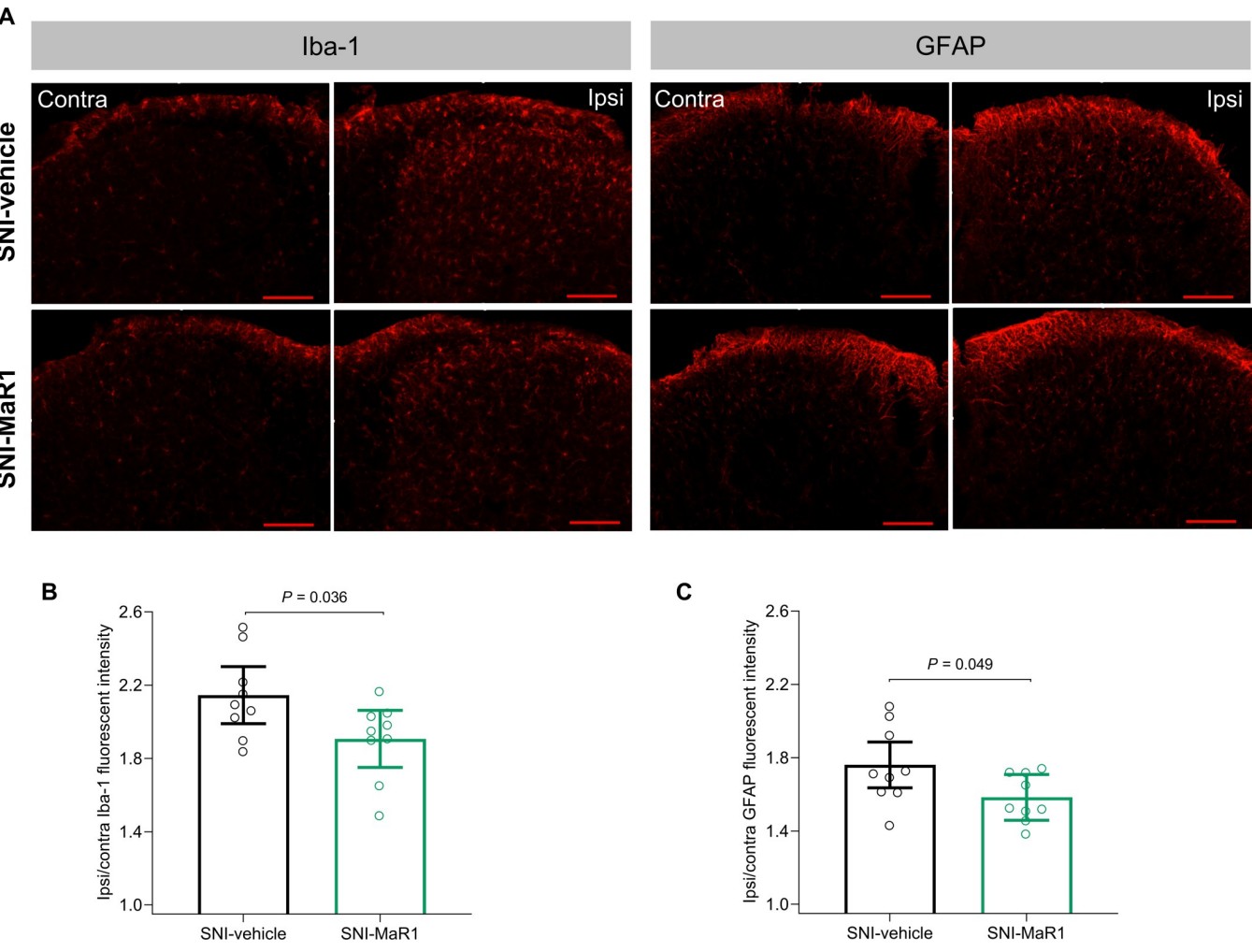

**Fig 4. MaR1 treatment reduced spared nerve injury (SNI)-induced microglial and astrocytic activation on day 12.** Representative images of the immunofluorescence staining with Iba-1 (microglial marker) and GFAP (astrocytic marker) in the spinal cord of mice (scale bar: 100 μm) (**A**). Quantification of Iba-1 (**B**) and GFAP (**C**) staining intensities in the dorsal horn is represented as the ipsilateral/contralateral ratio of mean fluorescence intensity. Data were analyzed with single measures ANOVA, with factor *Treatment*. Holm's procedure was used to correct for multiple comparisons. Individual values and mean predicted values (95% CI) from the analysis are presented. *P*-values for statistically significant comparisons are indicated. SNI-veh, n = 9; SNI-MaR1, n = 9. As previously referred, a female mouse from the SNI-vehicle group did not develop mechanical hypersensitivity after SNI and was therefore excluded from the study. Additionally, a male SNI-MaR1 mouse was excluded from the immunofluorescence analysis due to processing-associated tissue damage.

## Effects of MaR1 treatment on spinal concentration of cytokines

We also assessed whether the spinal concentrations of pro- and anti-inflammatory cytokines were modulated by MaR1 treatment in SNI-mice.

No statistically significant differences between sexes or treatment groups were found on IL-6 (largest F = 3.13, *P* = 0.115; Fig 5B), IL-10 (largest F = 2.78, *P* = 0.134; Fig 5C) or IL-1β, although a marginal main effect of *Sex* was found in the latter cytokine (F(1, 8) = 5.25, *P* = 0.051; Fig 5A). Indeed, in SNI-vehicle groups, IL-1β values tended to be higher in females than in males. In addition, analysis of M-CSF values revealed main effects of both *Sex* (F(1, 8) = 10.25, *P* = 0.013) and *Treatment* (F(1, 8) = 6.43, *P* = 0.035) factors. M-CSF concentrations were higher in MaR1-treated SNI-males when compared to either their vehicle-treated controls (borderline *P*-value of 0.053) or MaR1-treated SNI-females (*P* = 0.035) (Fig 5D).

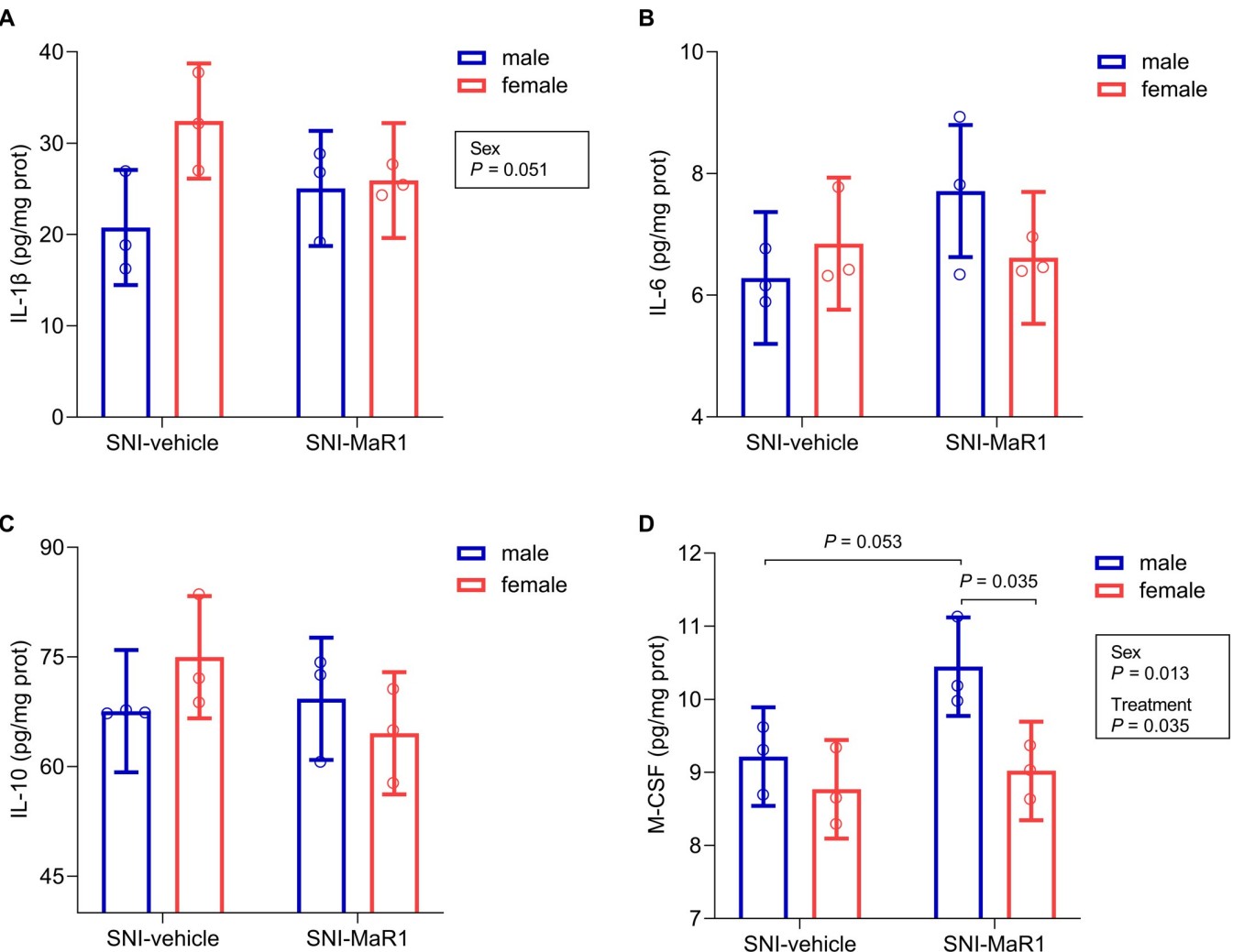

**Fig 5. Effects of MaR1 treatment on spinal inflammatory cytokines after spared nerve injury (SNI) on day 12.** MaR1 treatment did not influence the spinal concentrations of interleukin (IL)-1β **(A)**, IL-6 **(B)**, and IL-10 **(C)**, and increased macrophage colony-stimulating factor (M-CSF) values in males **(D)**. Data were analyzed with a single measures ANOVA, with factors *Treatment* and *Sex*. Holm's procedure was used to correct for multiple comparisons. Individual values and mean predicted values (95% CI) from the analysis are presented. *P*-values for statistically (or marginally) significant main effects/interactions and planned comparisons are indicated. ♂-SNI-veh, n = 3; ♂-SNI-MaR1, n = 3; ♀-SNI-veh, n = 3; ♀-SNI-MaR1, n = 3.

## Discussion

In the present study, oral treatment with MaR1 attenuated SNI-induced mechanical hypersensitivity and spinal glial activation in male and female mice. Furthermore, on day 11, male MaR1-treated mice appeared to show less escape/avoidance behavior than their vehicle-treated counterparts. Although not statistically significant, this difference is corroborated by more than one PEAP outcome measures used, and suggests MaR1 is able to ameliorate the affective component of pain in SNI-male mice. On the other hand, no definitive conclusions about the effects of MaR1 treatment on females could be drawn, since no increase in preference for the light side was detected in vehicle-treated SNI female mice, which by itself contradicts one of the fundamental assumptions of the PEAP test. No treatment effects were detected on spinal IL-1β, IL-6, and IL-10 values on day 12, but M-CSF concentrations were marginally higher in MaR1-treated SNI males.

## On the behavioral assessment of analgesic effects of MaR1 treatment through voluntary oral administration

Our results expand previously reported data showing antinociceptive actions of MaR1 treatment on different peripheral NP models. In most of the referred studies, MaR1 was intrathecally administered–non-compressive lumbar disk herniation (daily MaR1 administration during the first 3 postsurgical days; daily behavioral testing for mechanical and thermal hypersensitivity up to postsurgical day 7 [22, 23]); spinal nerve ligation (daily MaR1 administration from postsurgical days 3 to 5; daily hypersensitivity assessment up to postsurgical day 7 [20]); chronic constriction injury (single MaR1 injection 1 week after injury; hypersensitivity assessment up to 24h post-MaR1 dosing [21])–although there is also one publication, by Serhan *et al.*, reporting the antinociceptive effects of systemic MaR1 treatment in the vincristine-induced model of peripheral NP (i.p. injection of MaR1 shortly before administration of the chemotherapeutic agent vincristine reduced mechanical hypersensitivity between days 1 and 14 [15]). Our results not only further demonstrate the effectiveness of systemically administered MaR1 in a peripheral NP model, but specifically show for the first time that MaR1 administered through a voluntary oral protocol can ameliorate pain hypersensitivity in a NP model already installed, and maintain those antinociceptive effects for at least for six days beyond the treatment time window. Our results further add to the still scarce evidence showing that SPMs are orally bioactive, i.e. resist degradation in the acidic gastric milieu and reach therapeutic levels in plasma. LXs and aspirin-triggered analogs displayed potent anti-inflammatory actions when administered *ad libitum* via drinking water or through oral gavage [62, 63]. Likewise, oral RvD1 has demonstrated potent immunoresolving actions, and proved to elevate RvD1 plasma concentrations [64, 65], and to reduce the pro-inflammatory environment in the CNS, promoting an anti-inflammatory phenotype of microglia [66]. Recently, a series of papers by Moreno-Aliaga's group has described several beneficial actions of MaR1 administered by oral gavage in diet-induced obese mice [34–38], demonstrating that MaR1 is orally bioactive as well. The use of an oral route of administration, particularly by a non-stressful method, increases not only the translational potential of our findings but also their validity, since increased well-being and lower stress levels in the experimental animals produce better behavioral results [31].

**On the assessment of the affective component of pain: Effects of MaR1 treatment in the escape/avoidance behavior of SNI mice.**   Thus far, evaluation of SPMs' analgesic effects on peripheral NP models has almost exclusively relied on conventional reflexive pain measures, except for a study by Xu *et al.* showing that pretreatment with PD1 reduced spontaneous pain as assessed by a conditioned place preference test [12, 67]. Since pain is a complex and multidimensional experience, tests that only assess the sensory experience of pain do not reflect the global impact of antinociception. Thus, preclinical pain studies should also evaluate the affective component of pain [68, 69]. Operant learning assays, such as the PEAP test, have been proposed to test the motivational/aversive aspects of pain, thereby reflecting its emotional and unpleasant nature in humans [70, 71]. Our results suggest amelioration of the affective component of pain in MaR1-treated males and show that, when confronted with a choice, male C57BL/6J mice with SNI perform a purposeful behavior response to escape/avoid the noxious mechanical stimulation of the injured hind paw, on day 11 after surgery. Conversely, vehicle-treated SNI females do not develop preference for the light compartment throughout the test.

## On the applicability of the PEAP test in mice

The PEAP test has been increasingly used in rodent models of peripheral NP, generally revealing the development of an associated aversive behavior (e.g. [45, 47, 59–61]; but see also [56]).

However, one should keep in mind that conclusions drawn from the PEAP test may depend on methodological variables (e.g. frequency of stimulus application, force of the von Frey filament or test duration) [61], on the sex/strain/species tested, and on the outcome measures analyzed (e.g. total time in light or measures that take into consideration changes in behavior throughout the test period).

Most PEAP studies published so far have used rats. Considering the existence of behavioral differences between rats and mice that may influence their behavior in the PEAP test (e.g. mice are known to be more active and take longer to habituate to new environments than rats [72]), more studies are necessary to expand the validity of this test to NP mouse models as well. In this context, our findings add to the still scarce evidence of the applicability of the PEAP test in mice with peripheral NP. Using a protocol similar to ours, Grégoire *et al.* reported that SNI induced motivated pain avoidance in male CD1 mice, 7 months after surgery [45]. Other studies with mice have used modified protocol versions, including von Frey filaments ranging from low intensity to clearly suprathreshold (even to naïve animals) stimulus forces (0.07–6 g), and/or including unstimulated reference baselines (additional 5–10 min of free exploration periods at the beginning of the test) [59, 73, 74]. Santello *et al.* reported that male C57BL/6J mice switched their preference from the dark to the light compartment of the PEAP box, 7–8 days after CCI [74]. Using male and female subjects, Gan *et al.* reported a higher total time spent in light in C57BL/6J SNI-mice than in sham controls, on postsurgical days 14–16 [59, 73].

A clear lack of PEAP test results for female rodents have insofar precluded conclusions about the validity of the PEAP test for the assessment of the motivated pain avoidance behavior in females with peripheral NP. Indeed, ours is one of the first studies to evaluate the behavior of female rodents in this paradigm. Although Gan *et al.*, as already mentioned, have previously used both male and female mice, their results were pooled together and it is not clear whether differences or similarities between sexes were taken into consideration [59, 73]. Moreover, considerable methodological differences and the use of different measures as indicators of escape/avoidance behavior hamper direct comparisons with our study. By analyzing not only the total time in light, but also measures more focused on changes of behavior throughout the test period, our results showed that, under our conditions, the repetitive stimulation was not sufficiently aversive/unpleasant to push females out of the preferred dark side, raising the possibility that the PEAP test may not be appropriate to ascertain emotional pain behavior in female C57BL/6J mice with SNI-induced NP. Previous studies have associated lower force/frequency of mechanical stimulation with a decrease in the degree of escape/avoidance behavior in rats [61]. However, we used a suprathreshold von Frey filament in order to provide a consistently noxious stimulus. Therefore, we hypothesize that SNI effects on other non-specific behavioral responses, such as anxiety state, might have biased SNI females' behavior in the PEAP test. In fact, there is a clear resemblance between the PEAP and the light/dark box (LDB) test, which measures the anxiety-like behavior. In the LDB test, increased time spent in the light compartment and multiple transitions between chambers are suggested to reflect low levels of anxiety [75]. Curiously, we found a positive global correlation between these two parameters in the PEAP test, suggesting that they are indeed mutually related. Since more anxious individuals may spend more time in the dark side, false negatives may occur [44]. Nevertheless, the overall time spent in light did not differ between male and female vehicle-treated SNI mice in our study, and evidence of anxiety-like behavior following SNI has been inconsistent [76, 77], with some studies suggesting that nerve injury does not primarily induce anxiety-like behaviors in male C57BL/6 mice [78, 79], although studies with female mice are still lacking. Moreover, previous studies found no correlation between the PEAP behavior and anxiety levels in rodent pain models, suggesting that PEAP responses are not

confounded by the animals' state of anxiety [44, 48, 80]. Likewise, changes in locomotor activity do not appear to induce alterations in PEAP behavior [44], while differences in cognitive learning and/or memory impairment remains a potential contributor to individual variability, due to the potential influence on the learning process that is required in the PEAP test [48]. On the other hand, although our results, in line with the previous study by Grégoire *et al.* [45], validate the herein described PEAP protocol for male mice, the methodology may have to be adapted for female mice. For instance, Refsgaard *et al.* have validated the PEAP test in female C57BL/6J mice with inflammatory pain following a protocol with considerable differences, namely the application of a 4 g von Frey filament at 10-s intervals. Since mice display many transitions between chambers, the authors suggested that the development of spatiotemporal associations with the two different stimuli is hindered by the consequent short duration of each stay and introduced a 20-min training session that consisted of restricting each mouse for two 5-min periods in each of the two areas immediately before the test period [44]. Further studies to better understand the natural behavior of male and female mice in the PEAP testing apparatus (e.g. comparing the behavior of non-treated naïve and SNI animals on the PEAP box without mechanical stimulation), may enable optimization of test conditions.

## On the neuroinflammatory mechanisms underlying MaR1 actions

Mechanistically, we focused on neuroinflammation, specifically in the spinal cord. Systemically administered SPMs have been reported to act on the CNS, indicating that they are able to cross the blood-brain barrier, which is further supported by the fact that SPMs are small lipophilic molecules (similarly to what occurs for their precursors, DHA and EPA) [19, 81]. Also, intravenous or intraperitoneal treatments with MaR1 have resulted in attenuation of neuroinflammation, in models of spinal cord injury (SCI) [82], and experimental autoimmune encephalomyelitis (EAE) [83]. Specifically, in these models, systemically administered MaR1 has been shown to modulate several processes in the spinal cord, namely reduction of pro-inflammatory cytokines and immune cells counts, redirection of macrophage polarization towards an anti-inflammatory phenotype, and silencing of major inflammatory intracellular signaling cascades (e.g. STAT1, STAT3, STAT5, p38, ERK1/2) [82, 83]. Furthermore, in NP models, MaR1 attenuated nociception through several spinal mechanisms, such as suppression of NF-κB activation, restoration of synaptic integrity, or inhibition of NLRP3 inflammasome-induced pyroptosis [20, 22, 23]. Moreover, MaR1 has been shown to inhibit NP-induced spinal microglial and astrocytic activation, as assessed by analysis of Iba-1 and GFAP immunoreactivity, either after intrathecal [20] or local application to the nerve [21]. Our results corroborate the ability of MaR1 to attenuate both microglial and astrocytic activation in NP models, and extend it to the oral administration route. Nevertheless, it cannot be concluded whether MaR1 effects on glial cells result from direct or indirect actions. Curiously, a previous study has reported that minocycline–widely used as a suppressor of microglial activation–improved the affective dimension of pain in NP patients [84]. These results suggest a possible role of MaR1 effects on microglia on the apparent amelioration of the escape/avoidance behavior in MaR1--treated males.

It is widely accepted that activated glia produce and release numerous mediators, including proinflammatory cytokines, chemokines, and growth factors, which are able to modulate excitatory and inhibitory synaptic transmission, and further act on glia cells, sustaining neuroinflammation [85]. Therefore, one would expect the reduction of spinal glia activation to reduce spinal pro-inflammatory cytokines. Indeed, other authors have reported reduced spinal pro-inflammatory cytokines during (or up to 2–4 days after) intrathecal or local (to the nerve) treatment with MaR1, in peripheral NP models [20–23]. Similarly, attenuated spinal IL-1β and

IL-6 values have also been reported during systemic treatment with MaR1, in SCI and EAE models [82, 83]. We did not detect differences in spinal IL-1β and IL-6 values between vehicle- and MaR1-treated SNI mice. These results, although in apparent contradiction with previously reported data, may be due to the fact that cytokines were evaluated only 7 days after MaR1 treatment cessation. Although the antinociceptive effect of the MaR1 treatment between post-injury days 3 and 5 was still present on day 11, no treatment-associated reduction in the proinflammatory cytokines IL-1β and IL-6 could be detected at this later timepoint. However, the fact that we used the whole L3-L5 spinal cord segment (both ipsi- and contralateral parts) for the cytokines determination might have contributed as a "dilution" factor, rendering the detection of differences in protein concentrations between experimental groups more difficult. We did not detect any effects of MaR1 treatment in spinal IL-10 either, which is in line with previous studies in NP models [21, 82]. Indeed, MaR1 has been suggested to preferentially modulate pro-inflammatory, rather than anti-inflammatory, cytokines [82].

We have also detected higher spinal M-CSF levels in males than in females, and a male-specific increase in M-CSF levels upon MaR1 treatment. Mounting evidence has implicated injured sensory neuron-derived M-CSF in the modulation of NP via central actions on microglia [86]. However, pain-inducing effects of M-CSF are sexually dimorphic, since intrathecal administration of M-CSF is sufficient to induce mechanical hypersensitivity in male, but not female, mice [87, 88]. This effect is reportedly associated with similar microglial proliferation in both sexes, as assessed by Iba-1 immunoreactivity [88], while transcriptomic profiling and morphological analysis reveal robust microglial activation only in males [87]. Okubo *et al.* reported that early phase (from post-surgical 0 to 40h) intrathecal treatment with a M-CSF receptor inhibitor significantly reduced mechanical hypersensitivity in SNI male rats for up to post-surgical day 3, while a later treatment, starting only 12 days after SNI, had no effects on mechanical hypersensitivity [89], which suggests that spinal M-CSF/M-CSF receptor signaling may be important to the initiation but not the maintenance of NP [86]. Consistent with that finding, the effects of minocycline, a non-specific microglial inhibitor, also seem to be limited to the early phase of nerve injury-induced mechanical hypersensitivity [90, 91]. Therefore, we speculate that MaR1 administration reduced or did not alter M-CSF values, and a compensatory mechanism (or a rebound effect) might have been triggered in males after treatment completion. The slight increase of M-CSF values supposedly occurred only in a later phase and, thus, was not associated with changes in pain hypersensitivity. Curiously, we have recently observed a similar result in male SNI-mice treated with a selective inhibitor of the NADPH oxidase 2, an enzyme whose primary function is the production of reactive oxygen species (ROS) [92]. Indeed, mounting evidence has been revealing an important relationship between M-CSF signaling and oxidative status [93, 94]. Therefore, since MaR1 may modulate ROS production through several mechanisms [95–97], we speculate that the drug-induced increase in M-CSF values might be associated with ROS modulation. Nevertheless, further work is necessary to confirm this observation, and to ascertain potential underlying mechanisms.

## On other limitations of the study and future directions

Further mechanistic studies are warranted to explain the antinociceptive effects of MaR1 observed in this study. We focused on the inflammatory events in the SC, but, since a systemic administration route was used and MaR1 was previously shown to contribute to NP pathophysiology by peripheral mechanisms as well, future studies should assess drug actions on the periphery. For instance, in a peripheral nerve injury model, MaR1 has shown to reduce the number of damaged DRG neurons, promote injured nerve regeneration (regulated neurite outgrowth through the PI3K–AKT–mTOR signaling pathway) and inhibit muscle atrophy

[21]. In addition, *in vitro* studies have shown that MaR1 inhibits TRPV1 currents in DRG neurons and promotes injured nerve regeneration [15, 21]. Considering that (i) DRG macrophages contribute to NP initiation and maintenance [98], (ii) MaR1 is synthesized by macrophages, and (iii) SPMs are both regulators of macrophage responses and effectors in macrophage-mediated responses [99], evaluation of the macrophage response in the DRG would also be relevant in the future. On the other hand, disclosing how systemically administered MaR1 induces changes in the spinal cord warrants further investigation. Other MaR1 doses and treatment schemes, including longer treatment protocols, should also be evaluated. Finally, as already mentioned, oral administration of MaR1 has previously demonstrated several beneficial actions on rodent models, and our results further suggest potential effects of MaR1 on the SNI model through this route. MaR1 may promote its own synthesis through enhancement of the MaR1/RORα/12-lipoxygenase circuit [100]. Indeed, in order to enable the pursuit of an oral administration route for MaR1, future studies should evaluate the pharmacodynamics and the pharmacokinetic profile of MaR1 after oral administration, as already reported for other SPMs [64].

## Conclusions

In conclusion, our study contributes to the knowledge about the therapeutic potential of MaR1 in peripheral NP, by including female subjects and evaluating both sensory and affective components of pain, upon MaR1 administration through voluntary oral intake. Our findings show that MaR1 attenuates SNI-induced mechanical hypersensitivity in both male and female mice. In males, MaR1 also appeared to ameliorate the affective component of pain, which has been understudied in this field. The therapeutic effects observed are associated with attenuated microglial and astrocytic activation in both sexes, and possibly involve modulation of M-CSF action in males. Moreover, our results suggest that the PEAP test may not be appropriate to evaluate the emotional pain behavior in female C57BL/6J mice with SNI-induced NP, underscoring the need to include females in preclinical pain studies.

## Supporting information

**S1 Fig. Individual mechanical withdrawal thresholds (MWTs, log-10 transformed values), assessed before surgery (baseline, BL), and on days 7 and 11 after spared nerve injury (SNI) surgery.** SNI-veh, n = 7; SNI-MaR1, n = 8.
(TIF)

**S2 Fig.** Individual behavior of male **(A)** and female **(B)** mice throughout the PEAP test period, on day 11 after spared nerve injury (SNI), presented as time spent in the light chamber (%) in time bins of 5 min throughout the 30-min test period. ♂-SNI-veh, n = 6; ♂-SNI-MaR1, n = 6; ♀-SNI-veh, n = 6; ♀-SNI-MaR1, n = 6.
(TIF)

**S3 Fig.** Individual behavior of male **(A)** and female **(B)** mice throughout the PEAP test period, on day 11 after spared nerve injury (SNI), presented as accumulated time spent in the light chamber (%) in time bins of 5 min. ♂-SNI-veh, n = 6; ♂-SNI-MaR1, n = 6; ♀-SNI-veh, n = 6; ♀-SNI-MaR1, n = 6.
(TIF)

**S1 File.**
(ZIP)

## Acknowledgments

The authors thank Ana Rita Martins (Arium, Sistemas de Diagnóstico, Lda, Lisboa, Portugal) for technical support in the Milliplex assay.

## Author Contributions

**Conceptualization:** Luísa Teixeira-Santos, Teresa Sousa, António Albino-Teixeira, Dora Pinho.

**Formal analysis:** Luísa Teixeira-Santos, Dora Pinho.

**Funding acquisition:** Luísa Teixeira-Santos, António Albino-Teixeira, Dora Pinho.

**Investigation:** Luísa Teixeira-Santos, Sandra Martins, Dora Pinho.

**Resources:** António Albino-Teixeira, Dora Pinho.

**Writing – original draft:** Luísa Teixeira-Santos.

**Writing – review & editing:** Luísa Teixeira-Santos, Teresa Sousa, António Albino-Teixeira, Dora Pinho.

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
