## [Decision Letter · Decision Letter 0]

20 Mar 2023

PONE-D-23-05673The pro-resolving lipid mediator Maresin 1 ameliorates pain responses and neuroinflammation in the spared nerve injury-induced neuropathic pain: a study in male and female micePLOS ONE

Dear Dr. Pinho,

Thank you for submitting your manuscript to PLOS ONE. After careful consideration, we feel that it has merit but does not fully meet PLOS ONE’s publication criteria as it currently stands. Therefore, we invite you to submit a revised version of the manuscript that addresses the points raised during the review process.

Based on the reviewers' suggestions, the paper needs major revision.  The reviewers' comments can be found below.

We look forward to receiving your revised manuscript.

Kind regards,

Tanja Grubić Kezele, Ph.D., M.D.

Academic Editor

PLOS ONE

Journal Requirements:

Reviewers' comments:

Reviewer's Responses to Questions

**Comments to the Author**

1. Is the manuscript technically sound, and do the data support the conclusions?

Reviewer #1: Partly

Reviewer #2: Partly

Reviewer #3: Yes

2. Has the statistical analysis been performed appropriately and rigorously? 

Reviewer #1: Yes

Reviewer #2: Yes

Reviewer #3: Yes

3. Have the authors made all data underlying the findings in their manuscript fully available?

Reviewer #1: Yes

Reviewer #2: Yes

Reviewer #3: Yes

4. Is the manuscript presented in an intelligible fashion and written in standard English?

Reviewer #1: Yes

Reviewer #2: Yes

Reviewer #3: Yes

5. Review Comments to the Author

Reviewer #1: The authors described the possible therapeutic potential of maresin-1 in spared nerve injury-induced neuropathic pain. I have one concern. In figure 5, please set the non-SNI control group to see if SNI treatment itself significantly increases these inflammatory cytokines, because previous many studies showed that maresin-1 suppresses the production of these representative inflammatory cytokines.

Reviewer #2: In this work, Teixeira-Santos and colleagues demonstrated that oral treatment with MaR1 is effective at reducing SNI-induced pain in rats. Novelty is the demonstration that voluntary intake of MaR1 in jam for 3 days is enough to reduce pain and mild sex-specific effect during escape/avoidance behaviors in MaR1-treated animals. The study is interesting, however, there are points that should be addressed:

1) How did Author monitor jam intake? Do all the animals ingest the whole 60 uL? Do Authors see any difference in intake for the vehicle- vs MaR1-containing jam? Would Authors expect gain weight upon chronic treatment?

2) In this work, the Authors focus on spinal cord changes upon SNI but performed systemic treatment with MaR1 rather than local via intrathecal injection. How do Authors propose MaR1 is reducing glial cell activation? Is this effect indirect or they propose that MaR1 is able to cross BBB? An explanation about how they envision peripheral treatment with MaR1 on the spinal changes is missing.

3) How do Authors explain increase in M-CSF induced by MaR1? Do they see resolution macrophages infiltrating the spinal cord upon treatment?

4) It is not clear to me the rationale for performing treatment only from day 3 to 5 (based on schematic provided in figure 1). While the Authors see that 3 days of treatment is enough to reduce pain long after treatment is done, they seem to not see major changes in the spinal cord. Do the Authors have data on the effect of longer treatment protocol with MaR1 (3 to 11, for example)?

5) Based on the 3-day treatment protocol, it would be interesting to see the effect of MaR1 on the spinal cord cytokine levels and glial cell markers at 7-day timepoint as well. If possible, it would be also interesting to see whether they see any difference in c-FOS is there, CGRP, and TRPV1 staining for example.

6) Fig 4: Do Authors see sex-specific modulation of the glial cell markers? Does MaR1 reduce that?

7) SNI often induces changes in toe spread. Do Authors see any difference in that parameter? Does MaR1 have an effect on that?

Reviewer #3: In the present manuscript, Teixeira-Santos et al studied the antinociceptive properties of MaR1 in a mouse model of SNI. Although several studies have already addressed this issue, the novelty of the present work relies of the fact that the authors have assessed whether the therapeutic properties of this bioactive lipid are sex dependent. Moreover, they also assess the effectivity of MaR1 when administered orally. They found that oral delivery of MaR1 attenuates pain responses in male and female mice and that these results were associated to the ability of this SPM in reducing astrocyte and microglia response in the dorsal horn of the spinal cord.

The experiments are well conducted, and the manuscript is interesting for the field.

Concerns

Did the author assess macrophage response in the DRG? This information would be of high interest.

Please show high magnification images of astrocytes and microglia

Please indicate whether error bars in the figures represent SEM or SD

6. PLOS authors have the option to publish the peer review history of their article (what does this mean?). If published, this will include your full peer review and any attached files.

Reviewer #1: No

Reviewer #2: No

Reviewer #3: **Yes: **Rubn López-Vales

---

## [Author Response · Author response to Decision Letter 0]

24 Apr 2023

Response to reviewers has been uploaded as a Word file.

---

## [Decision Letter · Decision Letter 1]

4 May 2023

PONE-D-23-05673R1The pro-resolving lipid mediator Maresin 1 ameliorates pain responses and neuroinflammation in the spared nerve injury-induced neuropathic pain: a study in male and female micePLOS ONE

Dear Dr. Pinho,

Thank you for submitting your manuscript to PLOS ONE. After careful consideration, we feel that it has merit but does not fully meet PLOS ONE’s publication criteria as it currently stands. Therefore, we invite you to submit a revised version of the manuscript that addresses the points raised during the review process.

Your manuscript, entitled "*The pro-resolving lipid mediator Maresin 1 ameliorates pain responses and neuroinflammation in the spared nerve injury-induced neuropathic pain: a study in male and female mice*", has been reviewed. Your efforts to revise the manuscript are appreciated. However, the peer review process continues because Reviewer 2 inquires more revision on certain issue that the author should address. Please find the reviewer's commentary below. 

We look forward to receiving your revised manuscript.

Kind regards,

Tanja Grubić Kezele, Ph.D., M.D.

Academic Editor

PLOS ONE

Reviewers' comments:

Reviewer's Responses to Questions

**Comments to the Author**

1. If the authors have adequately addressed your comments raised in a previous round of review and you feel that this manuscript is now acceptable for publication, you may indicate that here to bypass the “Comments to the Author” section, enter your conflict of interest statement in the “Confidential to Editor” section, and submit your "Accept" recommendation.

Reviewer #1: All comments have been addressed

Reviewer #2: (No Response)

2. Is the manuscript technically sound, and do the data support the conclusions?

Reviewer #1: Yes

Reviewer #2: Yes

3. Has the statistical analysis been performed appropriately and rigorously? 

Reviewer #1: Yes

Reviewer #2: Yes

4. Have the authors made all data underlying the findings in their manuscript fully available?

Reviewer #1: Yes

Reviewer #2: Yes

5. Is the manuscript presented in an intelligible fashion and written in standard English?

Reviewer #1: Yes

Reviewer #2: Yes

6. Review Comments to the Author

Reviewer #1: (No Response)

Reviewer #2: While the MS has improved after revision, this Reviewer understands that more mechanistic details are missing to explain the analgesic effect they observed. Also, data on the on the effect of the longer treatment protocol with MaR1 is required.

7. PLOS authors have the option to publish the peer review history of their article (what does this mean?). If published, this will include your full peer review and any attached files.

Reviewer #1: **Yes: **Yu Sawada

Reviewer #2: No

---

## [Author Response · Author response to Decision Letter 1]

29 May 2023

Response to Reviewers was uploaded as a separate Word file.

---

## [Decision Letter · Decision Letter 2]

5 Jun 2023

The pro-resolving lipid mediator Maresin 1 ameliorates pain responses and neuroinflammation in the spared nerve injury-induced neuropathic pain: a study in male and female mice

PONE-D-23-05673R2

Dear Dr. Pinho,

We’re pleased to inform you that your manuscript has been judged scientifically suitable for publication and will be formally accepted for publication once it meets all outstanding technical requirements.

Kind regards,

Tanja Grubić Kezele, Ph.D., M.D.

Academic Editor

PLOS ONE

Additional Editor Comments (optional):

Reviewers' comments:

Reviewer's Responses to Questions

**Comments to the Author**

1. If the authors have adequately addressed your comments raised in a previous round of review and you feel that this manuscript is now acceptable for publication, you may indicate that here to bypass the “Comments to the Author” section, enter your conflict of interest statement in the “Confidential to Editor” section, and submit your "Accept" recommendation.

Reviewer #1: All comments have been addressed

Reviewer #2: (No Response)

2. Is the manuscript technically sound, and do the data support the conclusions?

Reviewer #1: Yes

Reviewer #2: Yes

3. Has the statistical analysis been performed appropriately and rigorously? 

Reviewer #1: Yes

Reviewer #2: Yes

4. Have the authors made all data underlying the findings in their manuscript fully available?

Reviewer #1: Yes

Reviewer #2: Yes

5. Is the manuscript presented in an intelligible fashion and written in standard English?

Reviewer #1: Yes

Reviewer #2: Yes

6. Review Comments to the Author

Reviewer #1: They well responded to my all comments and I do not have any additional comments to this paper.

Reviewer #2: (No Response)

7. PLOS authors have the option to publish the peer review history of their article (what does this mean?). If published, this will include your full peer review and any attached files.

Reviewer #1: **Yes: **Yu Sawada

Reviewer #2: No

---

## [Editor Report · Acceptance letter]

13 Jun 2023

PONE-D-23-05673R2 

The pro-resolving lipid mediator Maresin 1 ameliorates pain responses and neuroinflammation in the spared nerve injury-induced neuropathic pain: a study in male and female mice 

Dear Dr. Pinho:

I'm pleased to inform you that your manuscript has been deemed suitable for publication in PLOS ONE. Congratulations! Your manuscript is now with our production department. 

Kind regards, 

on behalf of

Prof. dr. Tanja Grubić Kezele 

Academic Editor

PLOS ONE